# Enhancing COVID-19 Vaccination Awareness and Uptake in the Post-PHEIC Era: A Narrative Review of Physician-Level and System-Level Strategies

**DOI:** 10.3390/vaccines12091038

**Published:** 2024-09-11

**Authors:** Kay Choong See

**Affiliations:** Division of Respiratory and Critical Care Medicine, Department of Medicine, National University Hospital, Singapore 119074, Singapore; kaychoongsee@nus.edu.sg

**Keywords:** pandemic preparedness, public health, SARS-CoV-2, vaccination hesitancy, vaccines

## Abstract

Following the World Health Organization’s declaration that the COVID-19 pandemic is no longer a public health emergency of international concern (PHEIC), COVID-19 remains an ongoing threat to human health and healthcare systems. Vaccination plays a crucial role in reducing the disease’s incidence, mitigating its severity, and limiting transmission, contributing to long-term public health resilience. However, incomplete vaccination coverage and vaccine hesitancy exist. This narrative review investigates strategies at the system and physician levels aimed at sustaining awareness and uptake of COVID-19 vaccination in a post-PHEIC era. Through an examination of the existing literature, this review explores the effectiveness of diverse approaches utilized by healthcare systems and individual providers. These approaches address every component of the 5C model of vaccine hesitancy: confidence, complacency, constraints/convenience, calculation, and collective responsibility. Physician-level approaches include appropriate message framing, persuasive communication containing safety and personal/social benefit information, sharing of personal stories, creating a safe space for discussion, harnessing co-administration with annual influenza vaccines, and use of decision aids and visual messages. System-level approaches include messaging, mass media for health communication, on-site vaccine availability, pharmacist delivery, healthcare protocol integration, incentives, and chatbot use.

## 1. Introduction

Following the acute phase of the COVID-19 pandemic [1], the world is finally free from COVID-19’s pressure on healthcare systems. On 5 May 2023, the World Health Organization (WHO) declared that the COVID-19 pandemic was no longer a public health emergency of international concern (PHEIC). Nonetheless, post-PHEIC does not mean that the COVID-19 pandemic is over or that COVID-19 has become a benign disease. On the contrary, COVID-19 remains a threat to public health, and there is a need to ensure that the healthcare systems are not overwhelmed and that society does not face widespread disruption.

Vaccination plays a crucial role in reducing the incidence of COVID-19 disease, mitigating its severity, and limiting transmission. While COVID-19 vaccines have been instrumental in controlling the pandemic, new variants continue to cause surges in cases. Additionally, the effectiveness of earlier vaccines against new viral strains may decrease over time, emphasizing the need for continued vaccination efforts.

However, there are ongoing challenges to continued vaccination efforts, including incomplete vaccination coverage and reluctance among some individuals to get vaccinated. COVID-19 vaccination coverage is patchy even among previously vaccinated patients [2] who do not update their vaccination with variant-adapted boosters and risk becoming susceptible to infection again. In the post-PHEIC era, forceful public health interventions like mandatory vaccination have understandably been relaxed. Therefore, to enhance awareness and uptake of COVID-19 vaccines, other strategies are required. This review aims to describe the multiple non-mandatory physician-level and system-level COVID-19 vaccination strategies.

Such a review can equip physicians with evidence-based communication techniques to effectively engage with hesitant patients and inform system-level interventions that improve access and convenience. Physician-level strategies encompass personalized approaches tailored to individual patients and are implemented by frontline clinicians who engage directly with patients. Conversely, system-level strategies are designed for consistent application across all patients and individuals and are implemented at the organizational level. By synthesizing successful approaches, the review can guide policymakers in developing targeted equitable strategies that address public concerns, combat misinformation, and adapt to the evolving dynamics of the pandemic. This comprehensive understanding will help optimize vaccination efforts and strengthen preparedness for future public health crises.

## 2. COVID-19 Epidemiology in the Post-PHEIC Era

The COVID-19 pandemic was initially marked by waves of infection by different strains, starting from the ancestral wild-type strain, followed closely by the alpha, beta, gamma, delta, and omicron variants. Despite the decrease in intensive care unit admission rates in the post-PHEIC era [3], COVID-19 continues to pose a high disease burden, especially for unvaccinated individuals and those incompletely vaccinated against newer strains, and among subgroups like older adults and pregnant women who are vulnerable to severe disease [4,5,6]. In a 28-day period ending on 7 July 2024, the WHO reported 145,226 cases and 2213 deaths (https://data.who.int/dashboards/covid19/, accessed on 7 July 2024); coverage for healthcare workers and people over 60 years old has been declining.

When patients require hospitalization, morbidity and mortality associated with COVID-19 are similar to influenza. In a study conducted between February 2022 and May 2023, 4734 of the adults aged 60 or older who were hospitalized with acute respiratory illness across 25 hospitals in 20 U.S. states were diagnosed with COVID-19, while 746 were diagnosed with influenza [7]. Both groups of patients had similar levels of supplemental oxygen (58.2% versus 65.8%), high flow nasal oxygen or non-invasive ventilation (11.7% versus 13.7%), invasive intensive care unit admission (17.3% versus 16.8%), and invasive mechanical ventilation or hospital mortality (10.2% versus 7.0%). Additionally, among survivors of COVID-19 infection, COVID-19 may lead to post-acute sequelae [8] (also known as “long COVID”), which causes varying degrees of functional impairment and loss.

The combination of new virus variants [9] and declining vaccination coverage increases future risk of new pandemic waves. New virus variants have been associated with increased transmissibility [10,11] and hence greater and faster spread of the infection. These new variants are also known to possess immune evasion, which means that prior infection or vaccination with older strains afford less protection to individuals [12,13,14,15,16,17,18]. For instance, reinfection rates of 3 to 13% occur in patients infected with older variants [19].

Given the reduction or removal of non-pharmaceutical interventions like physical distancing or mask-wearing, vaccination continues to be the primary method for protection. Fortunately, variant-adapted booster vaccination remains effective in reducing hospitalization, mortality, and morbidity. Full vaccination can prevent post-COVID conditions (long COVID), with a 32% reduction in the odds of getting long COVID [20]. A focus on improving variant-adapted booster coverage of high-risk populations can have both health and economic payoffs. For instance, in a relatively small country like Singapore (approximately 4 million residents), booster vaccination in the elderly and high-risk individuals with comorbid conditions was estimated to avoid 278,614 cases, 21,558 hospitalizations, 239 deaths, >USD 200 million in direct medical costs, and >USD 500 million in indirect medical costs over one year [21]. Greater benefits were estimated when vaccination was extended to the standard-risk population.

However, several factors have hindered vaccination uptake in the post-PHEIC era, including low perceived vulnerability to severe consequences of COVID-19 infection, concerns about efficacy and side effects, lack of trust, and logistical challenges. Annoyance over COVID-19 booster vaccination has increased in one U.S. survey from 21.5% to 39.7% [22]. COVID-19 vaccine hesitancy also seems to be increasing, with rates in China rising from 8.39% in 2021 to 29.72% in 2023, contributed by urban residency, presence of chronic comorbid conditions, and lack of trust in vaccine developers [23]. In concert with increased vaccine hesitancy, COVID-19 booster vaccination rates have dropped to 13.76% among rural participants and 10.99% among urban participants in 2023 [24].

Vaccine hesitancy seems to be a special problem of COVID-19 vaccination compared to non-COVID-19 vaccines. In an international survey of 23,000 individuals across 23 countries in North America, South America, Europe, Asia, and Africa, vaccination intention dropped from 87.9% in 2022 to 71.6% in 2023. In the post-PHEIC era, 60.8% of these individuals expressed a higher willingness to get non-COVID-19 vaccines compared to COVID-19 vaccines, as a direct result of their negative experiences during the earlier acute phase of the pandemic [25].

The theoretical underpinning of vaccine hesitancy encompasses a variety of psychological, sociocultural, and behavioral theories that elucidate reasons for individual or group reluctance to accept vaccines, despite their accessibility. Several theories provide valuable perspectives on the cognitive and emotional processes, as well as the social and environmental contexts that shape vaccine-related decisions. The 5C model of vaccine hesitancy provides a framework for understanding and addressing the various barriers to vaccination uptake in the post-PHEIC era [26]. While other models such as Acceptance and Commitment Therapy (ACT), Capability, Opportunity, Motivation, and Behavior (COM-B), Health Belief Model (HBM) [27], Theory of Planned Behavior (TPB) [28], Social Cognitive Model (SCT), Theory of Reasoned Action and Planned Behavior (TRA), Transtheoretical Model (TTM), and Socio-ecological Model (SEM) [29] have been explored in studying vaccine hesitancy, the 5C model was selected for this review due to its ability to encompass a wider range of factors directly influencing decision-making processes at both the physician and system levels.

The 5C model incorporates the WHO’s Strategic Advisory Group of Experts on Immunization’s 3C model [30], and includes the following [26]:Confidence: Lack of trust in vaccines and the system administering them;Complacency: Overly optimistic perception of personal health status and low perceived vulnerability to infection or severe disease;Constraints (convenience): Structural barriers and accessibility issues;Calculation: Need for extensive and detailed information;Collective responsibility: Willingness to protect others and achieve herd immunity, i.e., communal orientation.

In the following sections, physician-level and system-level strategies will be described and tabulated using the 5C model.

## 3. Physician-Level Strategies for Enhancing COVID-19 Vaccination Awareness and Uptake

Physician-level strategies encompass tailored approaches for different individuals (Table 1). These approaches are particularly valuable for frontline clinicians who directly interact with patients and aim to address patient concerns about safety, side effects, and vaccine misinformation. Physician-level strategies require clinician time to provide personal recommendations, perform counseling, and employ patient decision aids. To support clinicians, communication training in techniques such as motivational interviewing can further improve the efficiency of health communications and increase vaccination uptake by patients.

Physicians across various specialties should offer vaccine education and create a supportive environment for discussing vaccinations, especially to address patient concerns about safety, side effects, and vaccine misinformation [31]. For patients willing to receive non-COVID vaccines, arranging for co-administration with these vaccines can enhance COVID-19 vaccine uptake. Coadministration with the seasonal influenza vaccine is particularly suitable given similar annual vaccination schedules and preservation of efficacy and safety for both vaccines [32,33].

Even a recommendation from a physician can decrease COVID-19 vaccine hesitancy, and any form of COVID-19 vaccine communication can enhance vaccine acceptance [34]. Messaging to patients should be personalized and fear tactics and myth-busting approaches should be minimized [35]. In contrast, balanced, authentic, and transparent messages about vaccines, including both benefits and drawbacks, would generally be well-received [36]. Further discussions should acknowledge patient concerns in a safe space, with physicians adopting a non-judgmental and supportive attitude [31].

To maximize the impact of physician recommendations, the content should feature an explicit recommendation (e.g., “you are due for a COVID vaccine booster”) rather than a participatory one (e.g., “are you interested in getting a COVID vaccine booster?”) [37]. Individuals with strong hesitancy may be influenced by the provision of personal benefit information, as evidenced by a U.K. population-based randomized trial of vaccination messaging [38]. Furthermore, a survey of 1706 adults in the U.S. showed that physician recommendations emphasizing an appeal to altruism (pro-social “protect others” message) had an additional impact in addition to acknowledging concerns about vaccination safety and efficacy [37].

Physician counseling can be augmented by using a patient decision aid, such as a web-based risk tool that visually depicts the relative risks associated with COVID-19 vaccination and infection as compared to other common activities [39]. Clinicians may also want to prioritize the amount of time spent on patient education, minimizing counseling to those patients who express a high level of COVID-19 acceptance or hesitancy. More time can then be reserved for patients who are undecided but open to discussion [40].

Nudging patients through the stages of behavior change (pre-contemplation, contemplation, preparation, action, and maintenance) is also crucial. Motivating patients to get vaccinated, engaging in their care, invoking altruism (especially among younger persons [41]), and avoiding false promises are essential steps. Assessing their readiness to change, strengthening motivation and confidence, and using open questions to address their health-related goals and objectives can all contribute to successful vaccine communication.

Clinician communication training, such as in motivational interviewing, can be beneficial to improve the efficiency of health communications and increase vaccination uptake by patients. Utilizing an online learning module with virtual simulation games based on evidence-based principles can also be advantageous, such as one based on the PrOTCT Framework. This framework is based on established principles as well as motivational interviewing strategies: (1) presume the patient will vaccinate; (2) offer to share knowledge and personal experiences with vaccines once you have explored their stance using OARS (open questions, affirmation, reflective listening, and summarizing reflections); (3) tailor recommendations to address patients’ specific health concerns; and (4) talk through a specific plan for when and where to get vaccinated [42]. These simulation games aim to increase confidence and self-efficacy in health communication about COVID-19 vaccination [43].

**Table 1 vaccines-12-01038-t001:** Physician-level strategies for enhancing COVID-19 vaccination awareness and uptake.

Strategy	Study Type and Population	Intervention	Outcome	Author (Year) [Reference]
Strategies that Address Confidence				
Framing the message using positive framing rather than negative framing	RCT, 1222 adults in the U.K.	Positive framing (i.e., avoidance of an adverse event) versus negative framing (i.e., occurrence of an adverse event) in patient information leaflets for vaccines	Positive framing increased vaccination intention for an unfamiliar vaccine but reduced intention for a familiar vaccine	Barnes (2022) [44]
Framing the message using descriptive risk labels and comparison with risks associated with other activities	RCT, 8998 adults in the U.S. and U.K.	Different ways of framing and presenting vaccine side-effects	Adding a descriptive risk label and a comparison to motor vehicle mortality increased willingness to take the COVID-19 vaccine by a total of 6.1 percentage points	Sudharsanan (2022) [45]
Incorporating safety information of vaccine-hesitant individuals with pre-existing health conditions	RCT, 5784 adults in Malaysia	14 experimental arms with different online web-based messages promoting COVID-19 vaccination	Participants rated their intent to recommend the COVID-19 vaccine to healthy adults, the elderly, and those with pre-existing conditions. Persuasive communication with safety information increased recommendation intention by 4–8 percentage points.	Hing (2022) [46]
Sharing of personal stories by peer mobilizers, after receipt of vaccination	Observational study. 197 newly vaccinated individuals	Monetary incentives offered to vaccinated individuals (peer mobilizers) for referring their family and friends for COVID-19 vaccination	45% of peer mobilizers referred at least one person who subsequently came in for vaccination. Peers’ vaccination experience influenced uptake by referred individuals	Thanel (2024) [47]
**Strategies that Address Complacency**				
Provision of personal benefit information	RCT, 15,014 adults in the U.K.	Provision of a range of brief written statements, addressing collective benefit, personal benefit, seriousness of the pandemic, and safety concerns	Provision of personal benefit information reduced hesitancy more than provision of collective benefit information	Freeman (2021) [38]
Creating a safe space for vaccine discussion	Survey of 531 physicians in primary care, emergency medicine, critical care, hospital medicine, infectious disease	Supportive and non-judgmental communication between the physician and patient to address patients’ questions about vaccination and to counteract misinformation. Include the sharing of personal vaccine status and appeals to civic responsibility	Qualitatively, these strategies have been successfully used by physicians to promote vaccination	Melnikow (2024) [31]
**Strategies that Address Constraints**				
Coadministration with another vaccine	Cohort study of 7399 healthcare workers from an Italian hospital	Co-administration of COVID-19 and influenza vaccines	Co-administration increased vaccine acceptance by 38%	Pascucci (2023) [48]
**Strategies that Address Calculation**				
Patient decision aid for vaccine counselling	Cohort study, 1015 residents in the U.S.	Relative Risk Tool *, which is an interactive web app that allows users to compare the risks associated with various scenarios and COVID-19 infection	The Tool was more effective than the U.S. CDC website for changing risk perception and increasing vaccination intent (people who claimed they would “definitely not” be vaccinated decreased by 9 to 9.8 percentage points)	Byerley (2024) [39]
Communicate uncertainties	RCT, 328 adults from a U.K. research panel	Announcements containing certainty versus uncertainty about the effectiveness of COVID-19 vaccination to prevent infection	Participants who received announcements containing certainty reported greater loss of vaccination intention and trust after receiving conflicting information, compared to those receiving announcements containing uncertainty	Batteux (2022) [49]
Use of visual messages	Quasi-experimental, 470 adults in Nigeria	Visual messages on COVID-19 vaccination	Subjects exposed to visual messages on COVID-19 vaccination, compared to those not exposed, reported greater intention to make themselves available for vaccination	Ugwuoke (2021) [50]
**Strategies that Address Collective Responsibility**				
Provision of altruism-eliciting information	RCT, 1373 Canadians aged 20–39 years	Altruism-eliciting short animated video intervention versus a text-based intervention focused only on non-vaccine-related COVID-19 preventive health measures	Prosocial and altruistic messages increased COVID-19 vaccine uptake	Zhu (2022) [51]

* https://www.covidtaser.com/relativerisk; CDC: Centers for Disease Control and Prevention. RCT: Randomized controlled trial.

## 4. System-Level Strategies for Enhancing COVID-19 Vaccination Awareness and Uptake

The strategies outlined in Table 2 are known as system-level strategies. While physician-level strategies involve tailored approaches for individual patients and require direct interaction between clinicians and patients, system-level strategies are designed to be applied consistently to all patients and individuals. They can be implemented by healthcare professionals and non-clinical leaders of healthcare institutions or smaller practices. These strategies focus on using various enablers such as reminder/recall systems, standing orders, improving access, providing incentives, engaging community leaders, and ensuring public access to accurate information. Additionally, the use of technology, including artificial intelligence and chatbots, can further support the dissemination of information about COVID-19 vaccination.

By adopting these system-level strategies, healthcare quality can be measured through process-type indicators. However, it is important to note that relying solely on system-level strategies may not effectively address highly vaccine-hesitant populations [52]. Furthermore, not all strategies will be effective in every scenario. For instance, convenient onsite vaccination did not boost vaccination rates among high-risk obstetric patients who were skeptical about the safety of the vaccine [53].

Conversely, in certain cases, system-level strategies may be necessary if individual-level interventions prove to be ineffective. A study involving 40 primary care providers from 11 clinics in the U.S. revealed that establishing a good and trusting physician–patient relationship alone may not be sufficient to change the minds of highly vaccine-hesitant patients. For these individuals, adding practical system-level changes, such as ensuring on-site availability, could help increase vaccine uptake [40].

Enablers for increasing vaccine uptake include utilizing reminder/recall systems with persuasive health communication [54,55,56], establishing standing orders for both inpatient and ambulatory settings [57,58], improving access (e.g., organizing workplace vaccination clinics [59], establishing ad hoc vaccination outreach services [60], and empowering pharmacists to vaccinate patients [61]), and providing incentives [62]. It is nonetheless important that reward systems are designed in a way that does not unfairly favor late adopters of vaccination [63]. Engaging trusted community leaders and vaccine ambassadors and ensuring public access to accurate information is also essential for combating misinformation, particularly on social media [64]. With the widespread availability of artificial intelligence technology and increasing public familiarity with chatbots, these tools can be leveraged to engage the public about COVID-19 vaccination [65,66]. They can facilitate preliminary discussions on the pros and cons of vaccination, thereby saving clinicians’ time and serving as a multiple-level healthcare communication approach.

**Table 2 vaccines-12-01038-t002:** System-level strategies for enhancing COVID-19 vaccination awareness and uptake.

Strategy	Study Type and Population	Intervention	Outcome	Author (Year) [Reference]
Strategies that Address Confidence				
Personal messages, e.g., short messages, phone call, e-mail, ad conventional mail	RCT. 862 adults in Mozambique	Phone calls with different message content	Vaccine acceptance was highest when the phone call included a vaccine endorsement, tapped into social memory of successful national vaccination campaigns, and used structured interaction to help participants develop a critical view against misinformation	Armand (2024) [67]
**Strategies that Address Complacency**				
Reminder messages, e.g., short messages, phone call, e-mail, and conventional mail	Cohort study. 3500 patients aged ≥65 years at Taipei Veterans General Hospital	BOOST (Booster promotion for older outpatients using SMS text reminders) program, with personalized reminders a week prior to a patient’s scheduled vaccination appointment	38% COVID-19 booster vaccination rate, compared to the 4% booster vaccination rate nationally	Lee (2024) [54]
Reminder messages using text messaging	RCT. 3,662,548 patients deemed eligible for COVID-19 booster vaccination in the U.S.	Reminder messages included communicating high current infection rates in a patient’s community and to plan for a specific date/time for vaccination	COVID-19 vaccination reminders increased the 30-day COVID-19 booster uptake by 21% (1.05 percentage points). Free round-trip rides to vaccination sites had no further benefit	Milkman (2024) [56]
Persuasive health communication. May be delivered using mass media, videos, and posters	RCT. 734 participants aged ≥18 years in Nigeria	Social media campaign delivered via Facebook	1.3 to 6 percentage point increase in COVID-19 vaccination	Evans (2024) [55]
Persuasive health communication. May be delivered using mass media, videos, and posters	Pre-post quasi-experimental study. 5804 survey respondents in Tanzania	The One by One: Target COVID-19 social media campaign by trained high-profile and high-impact influencers on the Twitter, Instagram, and Facebook platforms	1.7 percentage point increase in COVID-19 vaccination uptake among young adults aged 25–34 years, and 15 percentage point decrease in vaccine hesitancy among adults aged ≥35 years.	Kim (2024) [68]
**Strategies that Address Constraints**				
Enhanced vaccine access. e.g., make vaccines available on site	Cohort study of a hospital in South Africa	Onsite vaccination at the hospital for ambulatory patients	65% of 229 previously vaccinated patients opted for an additional dose on the day it was offered	Nair (2024) [59]
Vaccination by pharmacists	Cohort study of a Pharmacy COVID Champion Service in the U.K.	Community pharmacists empowered to counsel and to vaccinate patients	Between July and October 2021, among 8539 patients, 6094 patients agreed to vaccination, with 2019 initially hesitant patients converted	Micallef (2022) [61]
Ad hoc services for nursing homes, prisons, and homeless people	Cohort study of the Maximizing Uptake Program in the U.K.	Focused outreach activities (e.g., pop up clinics), engagement and communication for underrepresented groups (homeless, minority ethnic, refugees, asylum seekers, itinerant people, and physically/mentally impaired)	Vaccination of 7979 people from February–August 2021	Berrou (2022) [60]
Making vaccination part of healthcare protocols	Cohort study at Guy’s and St Thomas’ NHS Foundation Trust, U.K.	Inpatient COVID-19 vaccination protocol, i.e., opportunistic vaccination for hospitalized patients	34 patients were vaccinated in less than 2 months (November 2021 to January 2022, before protocol implementation), compared to 20 inpatients in 4 months (July 2021 to November 2021, after protocol implementation)	Bawa (2022) [57]
Making vaccination part of healthcare protocols	Quasi-experimental before-and-after study at the Atlanta Veterans Affairs Medical Center, U.S.	Inpatient COVID-19 vaccination protocol, i.e., opportunistic vaccination for hospitalized patients	COVID-19 vaccination rates during hospitalization increased significantly from 2% (16 of 769 inpatients) to 8% (61 of 793 inpatients) after protocol implementation	Fujita (2024) [58]
Financial and non-financial (e.g., awards) incentives for vaccine providers	Cohort study of Ohio’s Vax-A-Million initiative	Conditional cash lottery	Increased vaccinated proportion in the state by 1.5%, leading to significant reductions in COVID-19	Barber (2022) [62]
**Strategies that Address Calculation**				
Chatbot	Cohort study. 701 participants in France	Chatbot that answers questions about COVID-19 vaccines	Interacting with the chatbot for a few minutes significantly increased vaccination intention and positive attitudes toward COVID-19 vaccines	Altay (2023) [65]
Chatbot	Pre-post study. 46 participants in Hong Kong	Chatbot for promoting COVID-19 vaccination	Interacting with the chatbot significantly reduced hesitancy and increased vaccination intention	Luk (2022) [66]
**Strategies that Address Collective Responsibility**				
Persuasive health communication. May be delivered using mass media, videos and posters	Systematic review of 47 RCTs	Content of health communication	The most effective health communication strategies include providing information on vaccine safety and efficacy, using collective appeals combined with embarrassment appeals (i.e., evoking the emotion of embarrassment for non-vaccination), and featuring endorsements from political leaders	Xia (2024) [34]

RCT: Randomized controlled trial.

## 5. Limitations of This Review

While the review provides a comprehensive overview of various interventions by integrating findings from diverse study types and disciplines, it is important to acknowledge certain limitations. As this is a narrative review, the articles were not systematically selected and could encompass any paper describing the outcomes of physician or system-level interventions aimed at improving vaccine acceptance. At the same time, some relevant studies may have been excluded. Additionally, there may be potential bias and subjectivity in study selection and interpretation. The classification of the various strategies using the 5C model of vaccine hesitancy is also subjective. Nonetheless, the narrative review’s flexible and interpretive synthesis of the literature allows for the illumination of emerging trends, theoretical developments, and conceptual insights.

Furthermore, the studies featured in this narrative review were conducted across diverse populations and settings, making it a challenge to perform comparative effectiveness analysis of the different interventions. Consequently, future research could focus on comparing these interventions, potentially starting with those targeting the same domain (one of the 5Cs) and at the same level of intervention (e.g., physician-level or system-level).

## 6. Future Directions

Improving COVID-19 vaccination uptake may benefit from a whole-of-society approach [69,70], which involves integrating innovation, procurement, logistics, training, data and surveillance, access, and awareness. The role of artificial intelligence and chatbots can significantly contribute to this approach. It is essential to engage the public through in-person or online approaches. Providing real-time information on disease prevalence, vaccine efficacy, and complications to the public is crucial. Additionally, offering access to vaccination counseling and decision aids, as well as educating the public about vaccine safety surveillance systems [71], is imperative.

Addressing misinformation from social media and acknowledging concerns about unknown side effects are vital in combating vaccine hesitancy. Trustworthy information from healthcare professionals and government sources plays a significant role in increasing acceptance rates of COVID-19 vaccines [72,73]. Establishing a knowledge database that provides reliable, evidence-based, and freely accessible resources for both patients and providers is essential. Such platforms could use artificial intelligence techniques to collate real-time evidence-based information, such as papers demonstrating a lack of association between COVID-19 vaccination and stillbirth [74].

Finally, it is important to note that physician-level and system-level facilitators for vaccine awareness and uptake may vary among different participant groups, a consideration that future studies should explore. For example, individuals with post-COVID-19 physiological or psychological complications, including those diagnosed with long COVID, may exhibit reduced hesitancy toward receiving booster vaccination to prevent recurring infection. For these individuals, addressing concerns such as fear of vaccine side effects [75] and minimizing logistical and time-related barriers to vaccination should be primary areas of focus [76].

## 7. Conclusions

Despite the WHO’s declaration that the COVID-19 pandemic is no longer a public health emergency of international concern, the virus continues to pose a significant threat to public health. Vaccination remains crucial in reducing the spread and severity of the disease. To tackle challenges such as incomplete vaccination coverage and vaccine hesitancy, various physician-level and system-level strategies have been implemented. These strategies aim to improve access to vaccines, combat misinformation, and enhance vaccine uptake. Finally, exploring new methods of vaccine administration and developing vaccines that target a broader range of viral strains may further help improve vaccine acceptability and mitigate the ongoing threat of COVID-19.

## Data Availability

All data used can be found in the text and tables.

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
