# Peer review of "Enhancing COVID-19 Vaccination Awareness and Uptake in the Post-PHEIC Era: A Narrative Review of Physician-Level and System-Level Strategies"

_vaccines, 2024, doi:10.3390/vaccines12091038_

Round 1

Reviewer 1 Report

Comments and Suggestions for Authors

The manuscript was interesting but requires some content and editorial revision to enhance clarity and flow. See comments below.

Abstract:  It was generally clear.  Consider changing lines 23-25 more concise.

Consider changing “Systems-level approaches include personal and reminder messaging, appropriate use of mass media for persuasive health communication, making 23 vaccines available on-site and at ad hoc locations, such as pharmacies, and making vaccinations part of healthcare protocols, providing patient incentives, and utilizing chatbots.

Introduction:

Line 30: Remove “intense” from “intense pressure.”

Line 52: Consider changing  “the multiple non-mandatory physician-level and system-level strategies which have been implemented for COVID-19 vaccination” to “the multiple non-mandatory physician-level and system-level COVID-19 vaccination strategies.

Line 54: Remove “huge” from “huge waves.”

Line 64-65: Consider changing “When patients are ill enough to be hospitalized, morbidity and mortality are like an other well-known respiratory virus, influenza” to “When patients require hospitalization, morbidity and mortality associated with COVID-19 are similar to influenza.”

Lines 65-67: To enhance clarity and conciseness, please revise the statement beginning with “Among adults aged ≥60 years hospitalized….”

Lines 75-76:  Consider changing “conspire to increase future risk” to “increase future risk” and “fresh” to “new.”

Line 90: Consider changing “in only the elderly” to “in the elderly….”

Line 97:  Consider defining “low perceived vulnerability.”

Lines 102-3: Consider changing “low trust in vaccine developers” to “lack of trust with vaccine developers.”

Lines 109-112:   Consider changing “these individuals expressed being more willing to get non-COVID-19 vaccines compared to COVID-19 vaccines” to “these individuals expressed a higher willingness to get non-COVID-19 vaccines compared to COVID-19 vaccines.”

Lines 133-134:  Consider changing “already convinced for the need to vaccinate with non-COVID-19 vaccines” to “willing to receive non-COVID-vaccines….”

Line 140: Consider changing a “brief recommendation” to a “recommendation.”

Lines 142-144:  To increase conciseness and professional tone, consider revising the statement beginning with: “Messaging should be personalized for patients and should not rely on one- size-fits-all approaches containing fear appeals and myth-busting, which may have been more relevant during the acute phase of the pandemic but are less relevant after.”   One suggestion: “Messaging to patients should be personalized and fear tactics and myth-busting approaches should be minimized.”

Lines 144-146: Please consider revising the statement beginning with “Messages that are balanced…” to enhanced conciseness.

Lines 162-164: Consider revising the statement to enhance clarity and conciseness: “Busy clinicians may also want to tailor the amount of time spent on counselling, minimizing it for patients at either end of the vaccine acceptance/hesitancy spectrum” Suggested revision:

“Clinicians may also want to prioritize the amount of time spent on patient education, minimizing counseling to those patients who express a high level of COVID-19 acceptance or hesitancy.”

Line 169-170: Consider changing  “engaging with them in their care” to engaging in their care”  and “avoid making false promises” to “avoid false promises….”

Line 176: Consider changing  “for to improve efficiency” to “to improve efficiency.”

Line 179-180: Consider changing “PrOTCT Framework, which is in turn based on established principles of presumptive statements” to “PrOTCT Framework.  This framework is based on established principles as well as….”

Lines 199: Consider changing “who were sceptical about the vaccine's safety” to “who were skeptical about the safety of the vaccine….”

Line 220: Consider changing “as a system-level force multiplier for health communications” to “as a multiple-level healthcare communication approach.”

Line 227:  Consider changing “It is essential to engage through various channels, whether in person or online, to effectively reach the public” to “It is essential to engage the public through in person or online approaches.”

Line 229: Consider changing “disease seroprevalence” to “disease prevalence….”

Line 237: To avoid redundancy, consider changing “provides trustworthy, reliable” to “provides reliable….”

Line 240: Consider changing “no association between COVID-19 vaccination” to “lack of association between COVID-19 vaccination.”

Lines 242-252:  This paragraph does not transition well and seems extraneous. Consider deleting.

Pages 4-7, Table 1:

Within the outcome column under Sudharsanan et al, please make the outcome description more concise and clear.

Within the outcome column under Hing et al, please clarify what you mean by “improved recommendation intention.”

Within the intervention column under Thanel et al, please further define “peer mobilization” and peer referral” to enhance clarity and understanding of results.

Within the outcome column under Pascucci et al, please change the outcome results from “Co-administration increasingly accepted, rising 38% from” to “Co-administration increased vaccine acceptance to 38%”

Within the intervention column by Batteux et al, please further define  “certainty versus uncertainty about COVID-19 vaccination” to optimize clarity.

Page 8, Table 2:

Within the outcome column by Armand et al, please revise the results to increase conciseness and clarity.

Comments on the Quality of English Language

The manuscript was fairly clear but requires editorial revision because of syntax, grammatical, and spelling errors.  

Author Response

The manuscript was interesting but requires some content and editorial revision to enhance clarity and flow. See comments below.

[Reply] Thank you for your comments. I have reviewed the manuscript, updated the reference list by including Milkman’s megastudy (Nature 2024), and revised the text with the help of the reviewer comments.

Abstract:  It was generally clear.  Consider changing lines 23-25 more concise. Consider changing “Systems-level approaches include personal and reminder messaging, appropriate use of mass media for persuasive health communication, making 23 vaccines available on-site and at ad hoc locations, such as pharmacies, and making vaccinations part of healthcare protocols, providing patient incentives, and utilizing chatbots.

[Reply] Replaced “Systems-level approaches include personal and reminder messaging, appropriate use of mass media for persuasive health communication, making vaccines available on-site and at ad hoc locations, empowering pharmacists to deliver vaccination, making vaccination part of healthcare protocols, provision of incentives, and use of chatbots” with “System-level approaches include messaging, mass media for health communication, on-site vaccine availability, pharmacist delivery, healthcare protocol integration, incentives, and chatbot use.”

Introduction:

Line 30: Remove “intense” from “intense pressure.”

[Reply] Word removed as suggested.

Line 52: Consider changing  “the multiple non-mandatory physician-level and system-level strategies which have been implemented for COVID-19 vaccination” to “the multiple non-mandatory physician-level and system-level COVID-19 vaccination strategies.

[Reply] Change made as suggested.

Line 54: Remove “huge” from “huge waves.”

[Reply] Word removed as suggested.

Line 64-65: Consider changing “When patients are ill enough to be hospitalized, morbidity and mortality are like an other well-known respiratory virus, influenza” to “When patients require hospitalization, morbidity and mortality associated with COVID-19 are similar to influenza.”

[Reply] Change made as suggested.

Lines 65-67: To enhance clarity and conciseness, please revise the statement beginning with “Among adults aged ≥60 years hospitalized….”

[Reply] Replaced “Among adults aged ≥60 years hospitalized with acute respiratory illness prospectively enrolled from 25 hospitals in 20 U.S. states from February 2022 to May 2023, 4,734 had COVID-19 and 746 had influenza” with "In a study conducted between February 2022 and May 2023, 4,734 of the adults aged 60 or older who were hospitalized with acute respiratory illness across 25 hospitals in 20 U.S. states were diagnosed with COVID-19, while 746 were diagnosed with influenza."

Lines 75-76:  Consider changing “conspire to increase future risk” to “increase future risk” and “fresh” to “new.”

[Reply] Changes made as suggested.

Line 90: Consider changing “in only the elderly” to “in the elderly….”

[Reply] Word removed as suggested.

Line 97:  Consider defining “low perceived vulnerability.”

[Reply] Expanded the phrase to “low perceived vulnerability to severe consequences of COVID-19 infection.”

Lines 102-3: Consider changing “low trust in vaccine developers” to “lack of trust with vaccine developers.”

[Reply] Change made as suggested.

Lines 109-112:   Consider changing “these individuals expressed being more willing to get non-COVID-19 vaccines compared to COVID-19 vaccines” to “these individuals expressed a higher willingness to get non-COVID-19 vaccines compared to COVID-19 vaccines.”

[Reply] Change made as suggested.

Lines 133-134:  Consider changing “already convinced for the need to vaccinate with non-COVID-19 vaccines” to “willing to receive non-COVID-vaccines….”

[Reply] Change made as suggested.

Line 140: Consider changing a “brief recommendation” to a “recommendation.”

[Reply] Word removed as suggested.

Lines 142-144:  To increase conciseness and professional tone, consider revising the statement beginning with: “Messaging should be personalized for patients and should not rely on one- size-fits-all approaches containing fear appeals and myth-busting, which may have been more relevant during the acute phase of the pandemic but are less relevant after.”   One suggestion: “Messaging to patients should be personalized and fear tactics and myth-busting approaches should be minimized.”

[Reply] Change made as suggested.

Lines 144-146: Please consider revising the statement beginning with “Messages that are balanced…” to enhanced conciseness.

[Reply] Changed from “Messages that are balanced (incorporating data on both benefits and drawbacks), authentic and transparent about what is unknown about vaccines would generally be received favourably” to “In contrast, balanced, authentic, and transparent messages about vaccines, including both benefits and drawbacks, would generally be well-received.”

Lines 162-164: Consider revising the statement to enhance clarity and conciseness: “Busy clinicians may also want to tailor the amount of time spent on counselling, minimizing it for patients at either end of the vaccine acceptance/hesitancy spectrum” Suggested revision: “Clinicians may also want to prioritize the amount of time spent on patient education, minimizing counseling to those patients who express a high level of COVID-19 acceptance or hesitancy.”

[Reply] Change made as suggested.

Line 169-170: Consider changing  “engaging with them in their care” to engaging in their care”  and “avoid making false promises” to “avoid false promises….”

[Reply] Changes made as suggested.

Line 176: Consider changing  “for to improve efficiency” to “to improve efficiency.”

[Reply] Change made as suggested.

Line 179-180: Consider changing “PrOTCT Framework, which is in turn based on established principles of presumptive statements” to “PrOTCT Framework.  This framework is based on established principles as well as….”

[Reply] Changes made as suggested.

Lines 199: Consider changing “who were sceptical about the vaccine's safety” to “who were skeptical about the safety of the vaccine….”

[Reply] Change made as suggested.

Line 220: Consider changing “as a system-level force multiplier for health communications” to “as a multiple-level healthcare communication approach.”

[Reply] Change made as suggested.

Line 227:  Consider changing “It is essential to engage through various channels, whether in person or online, to effectively reach the public” to “It is essential to engage the public through in person or online approaches.”

[Reply] Change made as suggested.

Line 229: Consider changing “disease seroprevalence” to “disease prevalence….”

[Reply] Change made as suggested.

Line 237: To avoid redundancy, consider changing “provides trustworthy, reliable” to “provides reliable….”

[Reply] Change made as suggested.

Line 240: Consider changing “no association between COVID-19 vaccination” to “lack of association between COVID-19 vaccination.”

[Reply] Change made as suggested.

Lines 242-252:  This paragraph does not transition well and seems extraneous. Consider deleting.

[Reply] Paragraph deleted as suggested.

Pages 4-7, Table 1:

Within the outcome column under Sudharsanan et al, please make the outcome description more concise and clear.

[Reply] Replaced “Adding a descriptive risk label next to the numerical side-effect increased participants' willingness to take the COVID-19 vaccine by 3 percentage points. Adding a comparison to motor-vehicle mortality increased willingness by 2.4 percentage points. Combining both framing strategies increased willingness to receive the vaccine by 6.1 percentage points” with “Adding a descriptive risk label and a comparison to motor vehicle mortality increased willingness to take the COVID-19 vaccine by a total of 6.1 percentage points.”

Within the outcome column under Hing et al, please clarify what you mean by “improved recommendation intention.”

[Reply] Replaced “Persuasive communication containing safety information improved recommendation intentions to people with pre-existing health conditions by 4-8 percentage points” with “Participants rated their intent to recommend the COVID-19 vaccine to healthy adults, the elderly, and those with pre-existing conditions. Persuasive communication with safety information increased recommendation intention by 4-8 percentage points.”

Within the intervention column under Thanel et al, please further define “peer mobilization” and peer referral” to enhance clarity and understanding of results.

[Reply] Replaced “Peer referrals via peer mobilization with option of dis-tributing incentive coupons” with “Monetary incentives to offered to vaccinated individuals (peer mobilizers) for referring their family and friends for COVID-19 vaccination”.

Within the outcome column under Pascucci et al, please change the outcome results from “Co-administration increasingly accepted, rising 38% from” to “Co-administration increased vaccine acceptance to 38%”

[Reply] Change made as suggested.

Within the intervention column by Batteux et al, please further define  “certainty versus uncertainty about COVID-19 vaccination” to optimize clarity.

[Reply] Updated “Announcements containing certainty versus uncertainty about COVID-19 vaccination” to “Announcements containing certainty versus uncertainty about the effectiveness of COVID-19 vaccination to prevent infection”.

Page 8, Table 2:

Within the outcome column by Armand et al, please revise the results to increase conciseness and clarity.

[Reply] Replaced “Highest level of vaccine acceptance when the phone call contained an endorsement of the vaccine, activation of social memory of prior successful national vaccination campaigns, and using structured interaction to help the participant develop a critical view against misinformation” with “Vaccine acceptance was highest when the phone call included a vaccine endorsement, tapped into social memory of successful national vaccination campaigns, and used structured interaction to help participants develop a critical view against misinformation”.

Reviewer 2 Report

Comments and Suggestions for Authors

Ok...A somewhat interesting approach to a study.

My questions are 1) What is the importance of the study? There are so many COVID-19 studies what makes yours stand out? How does this narrative review assist with what we know now and what we should know? How will it lead to possible usage to improve services?

2) You need to discuss your findings more in-depth. Discuss more the similarities of the studies. You did a little for some them, but a narrative should tie together what you found

3) List the limitations to your narrative review. Since you alone did the review, there is a chance for bias, not locating/including studies that should been in the review, including studies that should not have been. You alone came up with the theoretical framework for the study. Issues that would be less problematic if you had other researchers. While narrative reviews may be conducted by one person, there is a greater need to avoid the aforementioned issues. 

Author Response

Ok...A somewhat interesting approach to a study.

My questions are 1) What is the importance of the study? There are so many COVID-19 studies what makes yours stand out? How does this narrative review assist with what we know now and what we should know? How will it lead to possible usage to improve services?

[Reply] Added to the Introduction: Such a review can equip physicians with evidence-based communication techniques to effectively engage with hesitant patients and inform system-level interventions that improve access and convenience. By synthesizing successful approaches, the review can guide policymakers in developing targeted, equitable strategies that address public concerns, combat misinformation, and adapt to the evolving dynamics of the pandemic. This comprehensive understanding will help optimize vaccination efforts and strengthen preparedness for future public health crises.”

2) You need to discuss your findings more in-depth. Discuss more the similarities of the studies. You did a little for some them, but a narrative should tie together what you found

[Reply] Expanded the first paragraphs of sections 3 (physician-level strategies) and 4 (system-level strategies) to tie the studies more.

3) List the limitations to your narrative review. Since you alone did the review, there is a chance for bias, not locating/including studies that should been in the review, including studies that should not have been. You alone came up with the theoretical framework for the study. Issues that would be less problematic if you had other researchers. While narrative reviews may be conducted by one person, there is a greater need to avoid the aforementioned issues.

[Reply] Included a new section 6 (Limitations of this review).

Reviewer 3 Report

Comments and Suggestions for Authors

General comments

This paper aims to review existing literature and explore the effectiveness of diverse approaches utilized by healthcare systems and individual providers. This is an important and interesting topic. I have some concerns and suggestions that may help improve the paper.

Major comments

  1. P3: This article discusses the 5C model of vaccine hesitancy and provides a framework; however, the authors should compare it with the other models or explain why this model is more suitable.
  2. Additionally, physician-level and system-level strategies should be considered different distinguishing criteria, and the authors should explain how to choose them.
  3. The content of this article mentions COVID-19 vaccine hesitancy, and it is suggested that a relevant theoretical basis can be put forward that makes it easier for the reader to understand
  4. This paper aims to review existing literature and explore the effectiveness of diverse approaches. What are the criteria for selecting review articles for this paper?
  5. In addition to vaccine hesitancy, many psychological traumas will also follow after the epidemic is over. It is recommended that the authors consider further research on long-term COVID.

Minor comments

  1. Separate the results, implications for managers, and future research into distinct sections to improve readability. Consider analyzing health literacy, government policies, and frameworks.

Author Response

General comments

This paper aims to review existing literature and explore the effectiveness of diverse approaches utilized by healthcare systems and individual providers. This is an important and interesting topic. I have some concerns and suggestions that may help improve the paper.

Major comments

P3: This article discusses the 5C model of vaccine hesitancy and provides a framework; however, the authors should compare it with the other models or explain why this model is more suitable.

[Reply] Added to the second last paragraph of section 2 (COVID-19 epidemiology in the post-PHEIC era): “While other models such as Acceptance and Commitment Therapy (ACT), Capability, Opportunity, Motivation, and Behaviour (COM-B), Health Belief Model (HBM), Theory of Planned Behaviour (TPB), Social Cognitive Model (SCT), Theory of Reasoned Action and Planned Behaviour (TRA), Transtheoretical Model (TTM), and Socio-ecological Model (SEM) have been explored in studying vaccine hesitancy, the 5C model was selected for this review due to its ability to encompass a wider range of factors directly influencing decision-making processes at both the physician and system levels”.

Additionally, physician-level and system-level strategies should be considered different distinguishing criteria, and the authors should explain how to choose them.

[Reply] Added to the last paragraph of section 1 (Introduction): “Physician-level strategies encompass personalized approaches tailored to individual patients and are implemented by frontline clinicians who engage directly with patients. Conversely, system-level strategies are designed for consistent application across all patients and individuals and are implemented at the organizational level”.

The content of this article mentions COVID-19 vaccine hesitancy, and it is suggested that a relevant theoretical basis can be put forward that makes it easier for the reader to understand

[Reply] Added to the second last paragraph of section 2 (COVID-19 epidemiology in the post-PHEIC era): “The theoretical underpinning of vaccine hesitancy encompasses a variety of psychological, sociocultural, and behavioural theories that elucidate reasons for individual or group reluctance to accept vaccines, despite their accessibility. Several theories provide valuable perspectives on the cognitive and emotional processes, as well as the social and environmental contexts, that shape vaccine-related decisions”.

This paper aims to review existing literature and explore the effectiveness of diverse approaches. What are the criteria for selecting review articles for this paper?

[Reply] As this is a narrative review, the articles were not systematically selected and could encompass any paper describing the outcomes of physician or system-level interventions aimed at improving vaccine acceptance. At the same time, some relevant studies may have been excluded. Additionally, there may be potential bias and subjectivity in study selection and interpretation. This has been included as a limitation in section 5.

In addition to vaccine hesitancy, many psychological traumas will also follow after the epidemic is over. It is recommended that the authors consider further research on long-term COVID.

[Reply] Added to section 6 (Future directions): “Finally, it is important to note that physician-level and system-level facilitators for vaccine awareness and uptake may vary among different participant groups, a consideration that future studies should explore. For example, individuals with post-COVID-19 physiological or psychological complications, including those diagnosed with long COVID, may exhibit reduced hesitancy towards receiving booster vaccination to prevent recurring infection. For these individuals, addressing concerns such as fear of vaccine side effects and minimizing logistical and time-related barriers to vaccination should be primary areas of focus”.

Reviewer 4 Report

Comments and Suggestions for Authors

This paper presents a descriptive review of prior studies of strategies at the system and physician levels aimed at sustaining awareness and uptake of COVID-19 vaccination in a post-public health emergency of international concern era. Overall, the paper is well written and the studies are well described. Here is a recommendation for revision.

The studies and their outcomes reported in Tables 1 and 2 are carefully described. This is good. In some cases, however, the studies that address one of the subsections of a table (e.g., "Strategies that address confidence" in Table 1) contain descriptions of more than one study. While recognizing the context-specific circumstances of these studies, it would be good for the author to include in the text some conditional statements about the relative effectiveness of the interventions. Among other things, this could be useful in guiding future studies.

Author Response

This paper presents a descriptive review of prior studies of strategies at the system and physician levels aimed at sustaining awareness and uptake of COVID-19 vaccination in a post-public health emergency of international concern era. Overall, the paper is well written and the studies are well described. Here is a recommendation for revision.

The studies and their outcomes reported in Tables 1 and 2 are carefully described. This is good. In some cases, however, the studies that address one of the subsections of a table (e.g., "Strategies that address confidence" in Table 1) contain descriptions of more than one study. While recognizing the context-specific circumstances of these studies, it would be good for the author to include in the text some conditional statements about the relative effectiveness of the interventions. Among other things, this could be useful in guiding future studies.

[Reply] The studies featured in this narrative review were conducted across diverse populations and settings, making it challenging to perform comparative effectiveness analysis of the different interventions. Consequently, future research could focus on comparing these interventions, potentially starting with those targeting the same domain (one of the 5Cs) and at the same level of intervention (e.g., physician-level or system-level). This limitation has been acknowledged in section 5 (Limitations of this review).

Round 2

Reviewer 2 Report

Comments and Suggestions for Authors

Ok...Looks like you addressed most of the concerns.

Reviewer 3 Report

Comments and Suggestions for Authors

After revising, this paper can be accepted.